# Do Families Exposed to Adverse Childhood Experiences Report Family Centered Care?

**Brianna M. Lombardi** [1,2,3,*], **Lisa d. Zerden** [2,3] 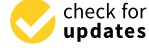, **Hyunji Lee** [4] and **Krissy Moehling Geffel** [5]

[1] Department of Family Medicine, School of Medicine, University of North Carolina at Chapel Hill, Chapel Hill, NC 27599, USA
[2] School of Social Work, University of North Carolina at Chapel Hill, Chapel Hill, NC 27599, USA
[3] The Cecil G. Sheps Center for Health Services Research, University of North Carolina at Chapel Hill, Chapel Hill, NC 27599, USA
[4] School of Social Work, University of Pittsburgh, Pittsburgh, PA 15213, USA
[5] Department of Family Medicine, School of Medicine, University of Pittsburgh, Pittsburgh, PA 15213, USA
[*] Correspondence: brianna_lombardi@med.unc.edu; Tel.: +1-984-974-4888

**Abstract:** Background: Youth from marginalized groups may be less likely to receive quality health care services. Adverse Childhood Experiences (ACEs) are known to impact long-term health, but it is unclear if there is a relationship between ACEs and receipt of Family Centered Care (FCC)—one indicator of high-quality health care. To assess this relationship, this study used a nationally representative sample of youth from the National Survey of Children's Health 2016–2017 combined data set. Caregivers of children who had at least one health care visit in the last 12 months (sub-sample n = 63,662) were asked about five indicators of FCC including if they felt the provider: (1) spent enough time, (2) listened carefully, (3) helped family feel like a partner, (4) provided information requested, and (5) showed sensitivity to culture. Methods: Logistic regression analyses examined the association between ACE score and each FCC quality indicator, as well between ACEs score and the overall FCC dichotomous score. Results: ACE exposure did not significantly predict access to a health care visit in the past 12 months. However, children with higher rates of ACEs were significantly less likely to receive FCC. Other factors that significantly predicted lower FCC included child race and ethnicity, insurance type, language in the home, and access to a regular health provider. Conclusions: Providers and health systems must identify, implement, and advocate for effective trauma-informed and care coordination interventions that ensure quality health care services for vulnerable children and families.

**Keywords:** family centered care (FCC); adverse childhood experiences (ACEs); health equity; trauma-informed care

## 1. Introduction

Children and families experiencing adverse childhood experiences (ACEs) are amongst the most vulnerable in our society. Indeed, ACEs are widely considered to be a major public health issue that impacts short and long-term health and well-being [1,2]. Although the association between early life adversity and morbidity is well documented, the pathways between these events are complicated [3]. Health care providers who specialize in the care and treatment of children, families, and adults play an important role in the prevention, intervention, and treatment of ACEs to mitigate long-term health impacts [4–6]. There has been extensive evidence documenting the negative effects of early adversity on child development and flourishing [7], further highlighting the importance of for early intervention across all systems that interact with youth and their families (i.e., educational, health and social service systems). Yet, few studies have investigated if children who experience ACEs report receipt of quality health care.

Advances in advocacy and policy in the 1990s promoted patient- and family-centered care (FCC) as one important component to quality health care. FCC has been described as "a partnership approach to health care decision-making" [8] (p. 298). FCC is deeply rooted in the belief that "the family is the child's primary source of strength and support and that the child's and family's perspective, and information are important in clinical decision making" [9] (p. 395). FCC shifted the focus from doctor directed health care to the inclusion and collaboration with patients and families. Several national and professional organizations, including the American Association of Pediatrics and the Maternal and Child Health Bureau, consider FCC to be best practice in pediatric health care clinics and for good reason. FCC is associated with positive health outcomes children, including increased parental satisfaction with services and a reduction in child emergency department visits [10–12]. The implementation and effectiveness of FCC is studied in many populations including with children with special health care needs [11], autism [13], and in early childhood outcomes [14].

Despite ample evidence that expanding FCC can support a child flourishing and help to reduce disparities in childhood health and well-being [14], FCC amongst families who are exposed to ACEs is not well-studied. Although FCC is associated with positive youth development and efficient family engagement in health services, not all children are recipients of FCC equally. Children from poor, uninsured, and publicly insured families are less likely to receive care consistent with FCC [15]. Relatedly, youth who are Black or Latinx are also less likely to receive FCC when compared to White youth [16,17], though findings in some studies are mixed [18].

As children who experience ACEs are more likely to be poor [19], uninsured or publicly insured, and of minority race or ethnicity [20], it is likely that ACE exposures may further complicate the influence of adversity on receipt of quality health care. However, as children with ACEs access pediatric care, this could be a setting where young people and families can receive the needed supports to promote health and well-being. Therefore, providing children with ACE exposure to quality care is critical. Using a nationally representative sample of children, this study assessed if children exposed to ACEs received FCC. Further, due to the strong evidence that children from racial, ethnic, and other minoritized groups are less likely to receive quality care, this study will focus on the multiple factors that may impact access to FCC.

## 2. Materials and Methods

### 2.1. Data Source

The data were drawn from the 2016 and 2017 combined sample of the National Survey of Children's Health (NSCH). The NSCH is a nationally representative dataset of children and youth in the United States collected by the U.S. Census Bureau. The NSCH uses a cross sectional study design to assess the physical and emotional wellbeing of children aged 0–17 years and has collected information on unique cohorts in 2003, 2007, 2011–2012, and the current sample in 2016 and 2017. Parents or guardians serve as respondents and provide responses for one child in their household randomly chosen as the survey subject. The 2016–2017 NSCH survey was administered electronically and by mail. Sampling weights were used to adjust for nonresponse and unequal selection bias. Using weighted data, results are representative of all noninstitutionalized children in the United States. A total of 71,811 surveys were completed and overall response rate was 41%.

### 2.2. Measures

FCC Indicators. Questions regarding FCC were asked to caregivers only if the child had a visit with a healthcare provider in the last 12 months. Five indicators of FCC were examined. Caregivers were asked if they felt the health care provider: (1) spent enough time spent with the child, (2) listened carefully to family, (3) helped family feel like a partner, (4) provides specific information requested, and (5) showed sensitivity to family's customs and values (culture). Respondents could answer Always, Usually, Sometimes, or

Never. This study utilized the NSCH created variable which dichotomized the Likert-type response options to Always/Usually (FCC = 1) or Sometimes/Never (FCC = 0) for analyses. A total FCC variable was used requires all five individual indicators of FCC to be Yes (1).

ACE Indicators. Nine variables measuring lifetime exposure to ACEs were used: extreme economic hardship, parental divorce/separation, parental incarceration, the child was witness to domestic violence in the home, the child was a victim/witness of neighborhood violence, the child lived with anyone with a drug or alcohol problem, the child lived with anyone with a mental illness or was suicidal, parent/guardian death, and the child was treated or judged unfairly due to race or ethnic group (discrimination). In the NSCH, all ACE variables were dichotomized, using a yes/no response, except for the question on economic hardship. This variable was first measured using a 4-point scale, ranging from 1 (never) to 4 (very often) and then dichotomized into a binary (yes/no) with yes describing very often and somewhat often. The NSCH is modeled after the original ACEs study [21] but diverges from the original items in several ways. First, the NSCH added questions on racial discrimination and exposure to community violence. Secondly, the NSCH did not assess child abuse and neglect. Third, the respondent is the caregiver instead of the individual themselves.

Covariates. Child and household covariates were selected and included in the model based on the prior literature. Child covariates included: Age (by year), race/ethnicity (Hispanic, White non-Hispanic, Black non-Hispanic, and multi-racial or other race/ethnicity non-Hispanic), gender (male/female), child insurance status (private, public, private, and public, uninsured), whether the child has regular health care provider for preventative and/or sick care (yes/no), and whether the child has special health care need (yes/no). One household covariate was included in the model: the language spoken in the home (English, Spanish, other language).

*2.3. Analysis*

Bivariate analyses examined the relationship between the number of ACEs reported and FCC indicators, as well as examined the difference of FCC reports between racial/ethnic groups. A chi-square bivariate analysis was used to test the significant relationships between the study variables at a significance level of 0.05. Five logistic regressions examined the relationship between ACE score and each FCC quality indicator adjusting for covariates, and one logistic regression between ACE score and the total FCC dichotomous score, controlling for all covariates. Interactions were tested and no significant interactions were found; as such, nonsignificant interactions were removed from the final model. Analyses were weighted to provide nationally representative estimates.

**3. Results**

The total sub-sample included 63,662 children who received a health care visit in the past year (85.6% of the total weighted NSCH sample). See Table 1 for the demographic description of the total NSCH, as well as the sub-sample of children who received a health care visit in the past year. The majority of the sub-sample was White (53.7%), followed by Latinx (22.8%), Black (13%), and multi-racial/other racial group (10.5%); 12.2% of the sub-sample lived in non-English-speaking households. Among the sub-sample, 44.9% experienced more than one ACE and close to 21% reported two or more ACEs. There were no significant differences between the number of ACEs and likelihood of receiving a health care visit in the past year ($p > 0.05$). However, children who were Black, Latinx, or other racial or ethnic background were less likely to receive a yearly health care visit than White children ($x^2 = 1033.65$, $p < 0.001$).

**Table 1.** Descriptive Statistics for Total Sample (N = 71,811) and Sub-sample of Children Who Received a Healthcare Visit in the last 12 months (N = 63,662).

| Variables | Total Sample | | Sub-Sample | | *p*-Value |
|---|---|---|---|---|---|
| | % | Unweighted *n* | % | Unweighted *n* | |
| Child received health care visit in the past year | | | 85.6 | 63,662 | |
| Child sex (male) | 51.1 | 36,800 | 51.2 | 32,581 | 0.579 |
| Child age (M, SE) | 8.61 (SE = 0.045) | 71, 811 | 8.40 (SE = 0.047) | 63,662 | <0.001 |
| Child race | | | | | <0.001 |
|   Hispanic | 24.7 | 7993 | 22.8 | 6708 | |
|   White, non-Hispanic | 51.4 | 50,219 | 53.7 | 45,261 | |
| Black, non-Hispanic | 13.1 | 4236 | 13.0 | 3694 | |
| Multi-racial/Other | 10.8 | 9363 | 10.5 | 7999 | |
| Household language | | | | | <0.001 |
|   English | 85.6 | 66,763 | 87.8 | 59,764 | |
|   Spanish | 9.5 | 2032 | 7.9 | 1525 | |
|   Other | 4.9 | 2498 | 4.3 | 1956 | |
| Health insurance status | | | | | <0.001 |
|   Public only | 31.5 | 13,391 | 30.8 | 11,698 | |
|   Private only | 57.6 | 52,109 | 60.3 | 47,050 | |
|   Public and private | 4.7 | 2625 | 4.8 | 2355 | |
|   Uninsured | 6.2 | 2624 | 4.1 | 1753 | |
| Special healthcare need | 18.8 | 16,304 | 20.4 | 15,476 | <0.001 |
| Usual sick or prevention care | 92.4 | 67,256 | 95.2 | 60,849 | <0.001 |
| Total ACE [1] count (M, SE) | 0.87 (SE = 0.011) | 70,825 | 0.87 (SE = 0.012) | 62,882 | 0.832 |
| Family Centered Care (FCC) | | | | | |
| FCC (always/usually) | | | 86.7 | 56,659 | |
| Components of FCC (always/usually) | | | | | |
|   Spend enough time with the child | | | 90.7 | 58,745 | |
|   Listen carefully | | | 95.0 | 60,322 | |
|   Sensitivity to your family's values and customs | | | 94.6 | 60,250 | |
|   Provide the specific information you needed | | | 95.2 | 60,323 | |
|   Help you feel like a partner in this child's care | | | 94.1 | 59,670 | |

[1] ACEs = adverse childhood experiences; FCC = family centered care; M = mean; SE = standard error. For age, children who received health care visit in the past year were younger than those who did not.

Table 2 presents the bivariate analyses of caregivers reporting overall FCC and receiving one of the five FCC indicators. Caregivers who reported zero ACEs were more likely to report always or usually experiencing all five FCC quality indicators, as compared to caregivers whose children experienced one or more ACEs (*p* < 0.001). For example, 96% of caregivers with a child of zero ACEs reported their health provider partnered with them in their child's care whereas only 88% of caregivers with a child of four or more ACEs felt this kind of provider-family partnership. Similarly, 97% of caregivers of a child with no ACE exposure reported their health provider always or usually listened to their concerns compared to 88% of caregivers with a child with four or more ACEs. Although 91% of caregivers with a child of no ACEs reported receiving overall FCC, only 76% of caregivers with a child of four or more ACEs received FCC.

Table 3 presents the findings for the six logistic regressions. Logistic regression analyses identified there was a significant relationship between the number of ACEs reported and the quality of health care received for each FCC indicator. Specifically, for children with four or more ACEs reported, the odds the caregiver reported receipt of total FCC was significantly lower compared to those with no ACE (OR = 0.41, 95% CI 0.32–0.52, *p* < 0.001). Significant differences also existed across each FCC indicator, with caregivers of children with four or more ACEs reporting lower odds of receiving each aspect of FCC compared to children with no ACE exposures: providing information (OR = 0.29, 95% CI 0.20–0.42,

$p < 0.001$); listening (OR = 0.25, 95% CI 0.17–0.36, $p < 0.001$); sensitivity to culture (OR = 0.34, 95% CI 0.29–0.43, $p < 0.001$); time provider spent with child/family (OR = 0.41, 95% CI 0.31–0.55, $p < 0.001$); and provider helped family feel like a partner (OR = 0.29, 95% CI 0.21–0.42, $p < 0.001$).

**Table 2.** Bivariate Analysis of Family Centered Care.

| Variables | FCC Yes (%) | Provide Info Always/Usually (%) | Listen Always/Usually (%) | Culture Always/Usually (%) | Time Always/Usually (%) | Partner Always/Usually (%) |
|---|---|---|---|---|---|---|
| Number of ACEs [2] | | | | | | |
| 0 | 90.5 | 97.0 | 97.0 | 96.6 | 93.5 | 96.2 |
| 1 | 84.7 | 95.0 | 94.3 | 94.2 | 89.2 | 93.5 |
| 2 | 81.2 | 92.3 | 92.1 | 90.7 | 86.7 | 90.8 |
| 3 | 78.5 | 91.9 | 90.7 | 90.3 | 84.9 | 88.7 |
| 4 or more | 76.2 | 88.3 | 88.1 | 88.8 | 83.3 | 87.6 |
| Child race | | | | | | |
| Hispanic | 80.8 | 93.4 | 93.3 | 92.1 | 84.6 | 92.2 |
| White, non-Hispanic | 91.0 | 96.6 | 96.3 | 96.6 | 94.5 | 95.6 |
| Black, non-Hispanic | 80.3 | 92.9 | 92.9 | 91.9 | 85.9 | 91.6 |
| Multi-racial/Other | 84.9 | 94.5 | 94.6 | 93.3 | 90.1 | 92.9 |
| Household language | | | | | | |
| English | 88.2 | 95.9 | 95.8 | 95.6 | 92.0 | 94.8 |
| Spanish | 74.4 | 90.2 | 89.6 | 87.2 | 78.6 | 88.8 |
| Other | 76.2 | 89.3 | 89.9 | 87.6 | 85.9 | 88.8 |
| Usual sick or prevention care | | | | | | |
| Yes | 87.6 | 95.7 | 95.6 | 95.2 | 91.6 | 94.6 |
| No | 69.3 | 85.3 | 85.4 | 84.7 | 74.3 | 83.3 |
| Special healthcare need | | | | | | |
| Yes | 82.4 | 92.0 | 92.7 | 92.4 | 88.6 | 91.5 |
| No | 87.8 | 96.0 | 95.6 | 95.2 | 91.3 | 94.7 |

[2] ACEs = adverse childhood experiences; *FCC* = family centered care.

**Table 3.** Logistic Regression Models Predicting Family Centered Care.

| Variables | FCC OR | FCC 95% CI | Info OR | Info 95% CI | Listen OR | Listen 95% CI |
|---|---|---|---|---|---|---|
| Number of ACEs [3] (ref: no ACE) | | | | | | |
| 1 | 0.68 *** | [0.58, 0.81] | 0.70 * | [0.52, 0.94] | 0.60 ** | [0.45, 0.80] |
| 2 | 0.55 *** | [0.44, 0.68] | 0.44 *** | [0.29, 0.66] | 0.37 *** | [0.26, 0.54] |
| 3 | 0.42 *** | [0.32, 0.55] | 0.39 *** | [0.25, 0.60] | 0.30 *** | [0.20, 0.45] |
| 4 or more | 0.41 *** | [0.32, 0.52] | 0.29 *** | [0.20, 0.42] | 0.25 *** | [0.17, 0.36] |
| Race (ref: White non-Hispanic) | | | | | | |
| Hispanic | 0.62 *** | [0.51, 0.76] | 0.86 | [0.63, 1.17] | 1.00 | [0.76, 1.33] |
| Black, non-Hispanic | 0.57 *** | [0.47, 0.69] | 0.78 | [0.60, 1.03] | 0.77 | [0.58, 1.03] |
| Multi-racial/Other | 0.71 *** | [0.60, 0.85] | 0.90 | [0.69, 1.19] | 1.02 | [0.76, 1.36] |
| Language (ref: English) | | | | | | |
| Spanish | 0.52 *** | [0.37, 0.72] | 0.41 ** | [0.24, 0.71] | 0.36 *** | [0.22, 0.61] |
| Other | 0.48 *** | [0.36, 0.63] | 0.40 *** | [0.26, 0.62] | 0.38 *** | [0.23, 0.62] |
| Special healthcare need (ref: No) | 0.76 *** | [0.66, 0.88] | 0.57 *** | [0.45, 0.73] | 0.63 *** | [0.50, 0.80] |
| Usual sick or prevention care (ref: Yes) | 0.44 *** | [0.33, 0.60] | 0.37 *** | [0.27, 0.52] | 0.40 *** | [0.28, 0.56] |
| Health insurance status (ref: Public only) | | | | | | |
| Private only | 1.30 ** | [1.12, 1.52] | 1.47 ** | [1.15, 1.88] | 1.27 | [1.00, 1.63] |
| Public and private | 0.64 ** | [0.46, 0.88] | 0.77 | [0.46, 1.31] | 0.63 | [0.39, 1.03] |
| Uninsured | 0.64 ** | [0.47, 0.86] | 0.50 ** | [0.34, 0.74] | 0.56 ** | [0.38, 0.83] |

**Table 3.** *Cont.*

| Variables | Culture | | Time | | Partner | |
|---|---|---|---|---|---|---|
| | OR | 95% CI | OR | 95% CI | OR | 95% CI |
| Number of ACEs (ref: no ACE) | | | | | | |
| 1 | 0.69 ** | [0.52, 0.91] | 0.69 *** | [0.56, 0.85] | 0.63 ** | [0.48, 0.82] |
| 2 | 0.41 *** | [0.29, 0.58] | 0.54 *** | [0.41, 0.70] | 0.39 *** | [0.28, 0.56] |
| 3 | 0.35 *** | [0.24, 0.51] | 0.42 *** | [0.30,0.58] | 0.29 *** | [0.20, 0.43] |
| 4 or more | 0.34 *** | [0.23, 0.49] | 0.41 *** | [0.31, 0.55] | 0.29 *** | [0.21, 0.42] |
| Race (ref: White non-Hispanic) | | | | | | |
| Hispanic | 0.81 | [0.61, 1.07] | 0.48 *** | [0.38, 0.61] | 0.89 | [0.65,1.22] |
| Black, non-Hispanic | 0.64 ** | [0.49, 0.82] | 0.48 *** | [0.39, 0.59] | 0.76 | [0.57, 1.03] |
| Multi-racial/Other | 0.73 * | [0.56, 0.95] | 0.67 *** | [0.54, 0.83] | 0.82 | [0.65, 1.05] |
| Language (ref: English) | | | | | | |
| Spanish | 0.34 *** | [0.21, 0.55] | 0.51 *** | [0.35, 0.73] | 0.40 ** | [0.24, 0.67] |
| Other | 0.38 *** | [0.25. 0.58] | 0.62 ** | [0.44, 0.86] | 0.45 *** | [0.30, 0.68] |
| Special healthcare need (ref: No) | 0.72 ** | [0.57, 0.92] | 0.83 | [0.69, 1.00] | 0.71 ** | [0.57, 0.90] |
| Usual sick or prevention care (ref: Yes) | 0.43 *** | [0.31, 0.60] | 0.38 *** | [0.27, 0.53] | 0.38 *** | [0.25, 0.59] |
| Health insurance status (ref: Public only) | | | | | | |
| Private only | 1.48 ** | [1.17, 1.86] | 1.28 * | [1.06, 1.54] | 1.04 | [0.83, 1.32] |
| Public and private | 0.85 | [0.52, 1.39] | 0.76 | [0.51, 1.12] | 0.72 | [0.40, 1.27] |
| Uninsured | 0.62 * | [0.42, 0.91] | 0.70 * | [0.49, 1.00] | 0.63 * | [0.41, 0.96] |

[3] ACEs adverse childhood experiences; FCC family centered care; OR odds ratio; CI confidence interval. Child's gender and age were controlled for in the logistic regression models, but were not significant at the 0.05 level. * $p < 0.05$, ** $p < 0.01$, *** $p < 0.001$.

In logistic regression analyses, race and ethnicity was not always a significant predictor of every individual FCC indicator. For example, there were no differences between White and Black respondents on the FCC indicators for partnering, listening, and providing information when controlling for other covariates. For Hispanic children, there was only a significant difference on the FCC indicator for time spent with the child during a medical visit with lower odds of receipt as compared to White children (OR = 0.48, 95% CI 0.38–0.61, $p < 0.001$).

Some of the most striking predictors of FCC in logistic regression analyses were the covariates: language spoken in the home, a place for usual or preventative care, the child having a special health care need, and the child's health insurance type. For example, non-English-speaking households had lower odds of receiving total FCC and each of the five FCC indicators, compared to English-speaking households (total FCC Spanish-speaking household: OR = 0.52, 95% CI 0.37–0.72, $p < 0.001$; total FCC other language-speaking household: OR = 0.48, 95% CI 0.36–0.63, $p < 0.001$). Further, it appears that having a regular health provider for usual or preventive care increases the likelihood of experiencing FCC (not having usual or preventative care OR = 0.44, 95% CI 0.33–0.60, $p < 0.001$). Finally, children with private health insurance had higher odds of receiving total FCC (OR = 1.30, 95% CI 1.12–1.52, $p < 0.01$) and quality FCC indicators for providing information (OR = 1.47, 95% CI 1.15–1.88, $p < 0.01$), sensitivity to culture (OR = 1.48, 95% CI 1.17–1.86, $p < 0.01$), and time provider spent with child/family (OR = 1.28, 95% CI 1.06–1.54, $p < 0.05$), compared to children who only had publicly funded health insurance.

## 4. Discussion

FCC is needed to provide children and families quality health care services [8]. Indeed, by providing services that incorporate FCC, children are more likely to have enhanced health outcomes through receiving early prevention, intervention, and treatment that promotes their health and well-being [10]. This study identified no differences in receipt of regular health care visits for children with ACEs compared to children with no ACE exposure. Recent work by Schweer-Collins and Lanier [7] acknowledged the association between higher ACEs, lower-quality care, and greater challenges for youth to access mental

health treatment—a finding we observed as well. Our results also revealed an inverse relationship between FCC and ACEs. Specifically, caregivers who reported higher ACE scores for the child also reported lower rates of FCC for both total FCC and across each FCC indicator. These findings highlight that the most vulnerable children and families may be receiving care that is reduced in quality. Most concerningly, there appears to be a dynamic impact in which children from marginalized groups, who are already at risk for ACEs, are also less likely to receive FCC. Study findings suggest intervention, system, and policy changes are needed to ensure equitable health care for all youth.

As hypothesized, families who experienced higher rates of ACEs were less likely to receive FCC even when controlling for race and ethnicity, source of usual care, language, and insurance status. In line with previous work, this finding suggests children with ACE exposure are uniquely at risk for receiving inferior quality of care. As children with ACEs have a higher risk of negative health outcomes and increased likelihood of needing referrals for mental health and other social services, this will require providers and health systems to engage in connecting families more purposefully to community and environmental supports to best meet their needs. This requires frequent and regular assessment for screening and intervention and referral to treatment needed to reduce risk for poor health trajectories. However, this also requires health systems to have integrated behavioral health supports and robust community-based supports to meet children and families' complex needs. Future work is needed to understand if FCC can promote appropriate service utilization and other positive health outcomes for families with ACEs, as FCC is demonstrated to be effective for children with special health care needs [11].

Beyond ACE exposure, this study highlighted several factors that predict inequality in FCC, including race and ethnicity, health insurance type, and language spoken in the home. It is not a new finding that children from marginalized backgrounds are less likely to receive quality health care. Numerous studies have identified that children who are Black, Latinx, and Asian experience reduced quality of care [22]. For example, racial and ethnic disparities exist in access to patient-centered medical care homes [23] and access to needed health services [24]. Building health systems to support the most vulnerable families is vital to addressing health equity.

This study particularly underscored that Latinx children and families who do not speak English in the home are at greatest risk for receiving fewer indicators of FCC; a finding that has been found within the existing literature [16]. Using data from the Medical Expenditure and Panel Survey and the National Health Interview, Latinx children were less likely to receive FCC related to provider communication as compared to white children [18]. Beyond Latinx ethnicity, findings from this study call out that families who do not primarily speak English have the lowest rate of total FCC and FCC indicators. Health systems must consider their ability to provide culturally and linguistically appropriate services for youth and families because it appears that this is a major driver for FCC [17]. As demographic populations shift in the U.S., meeting needs for Latinx families and Spanish-speaking families is imperative to providing FCC.

Like previous work, this study identified that the type of child health insurance is associated with the quality of health care services received [18]. In this study, caregivers of children with no insurance or public health insurance reported lower FCC as compared to those with private health insurance. Given the complexity of health insurance coverage in the U.S., this may exacerbate health disparities among children who already face multiple health and social risk factors. In 2018, 35% of U.S. children were covered through Medicaid or the Children's Health Insurance Programs (CHIP) while data from this same period also signaled a rise in uninsured children. Moreover, children of color, particularly Black and Latinx children are less likely to be covered by private insurance. This highlights major structural and systemic inequities that exist and are perpetuated related to race and ethnicity and health access and outcomes. Health equity for children cannot be achieved without a deep consideration of how racism manifests in how health care is accessed and

received. This reckoning will require a societal shift that conceptualizes health, and access to health care, as a human right [25].

*4.1. Future Directions for Research and Intervention*

Although screening for ACEs in pediatric settings is growing [26], more research is needed to assess the quality of care provided to children and families with high ACE exposure. Recent studies have identified the importance of roles of primary health care providers in screening for children at risk for ACEs in pediatric care settings [27] and positive attitudes associated with ACE screening tools [28]. However, most of this work pre-dates the COVID-19 pandemic which has exacerbated the mental health and social needs of youth and families. Additional research is needed to determine how knowledge of child ACE exposure impacts clinical practice. Finally, it is equally necessary to raise medical providers' awareness of ACEs and train all providers on how to offer appropriate services and referrals to high-risk families and children [26].

For families who experience ACEs, coordinating services, providing a trauma-informed framework, and utilizing FCC are all important components of care to improve the health and well-being of youth. Care coordination is an evidence-informed model that can enhance the quality of care and utilization of needed services for children and families who are most vulnerable [29]. Care coordination is patient-centered and aligns with the FCC approach [30]. Similarly, pediatric health clinics are encouraged to provide services within a trauma-informed framework. Although trauma-informed care is most often considered in behavioral health settings, pediatric and adult health providers are increasingly incorporating trauma-informed care strategies into their clinical practice [4,6,31]. Trauma-informed care includes patients and family engagement and involvement in their care, screening for trauma, training the workforce in trauma-specific treatment approaches, and engaging referral sources and partnering with cross-sector organizations [32]. Research suggests parents support a trauma-informed and person-centered approach to screening for ACEs in pediatric care settings [33]. Future research needs to integrate the necessary factors to provide quality health services for families exposed to ACEs. Likewise, medical providers, health systems, and professional organizations must advocate for policies that provide comprehensive care for children and families when ACEs are present [27,34].

*4.2. Limitations*

Study findings should be interpreted with possible limitations. ACE exposure was reported by the caregiver and could be higher or lower than stated. Further, the NSCH ACEs do not include child maltreatment of physical or sexual abuse, or neglect. Future studies should examine the influence of these adversities related to FCC and quality care. In addition, future work should examine the relative influence of individual ACE type by FCC. Similarly, FCC is reported by the caregiver, it is unknown if the child reports similar experiences with their care provider and previous studies report youth views are important to consider in quality-of-care research [35]. Studies that focus on a child's perspective of engaging with a health provider are needed. Analyses are cross-sectional which limits the ability for understanding causality between the measured constructs. Future longitudinal studies should examine how early life engagement in health care services impacts long term health for individuals exposed to ACEs—something that may have been impacted most recently by disruptions caused by COVID-19. This study did not closely examine the experience of Asian American or children of Asian immigrants and future work is needed to understand culturally responsive FCC for all youth.

**5. Conclusions**

This study identified that increased ACE exposure resulted in lower odds of receipt of FCC. In addition, African American and Latinx families, as well as families from other marginalized groups, were significantly less likely than white families to receive FCC. Non-English-speaking families, children who were uninsured, and families without a

regular health care provider reported the lowest rates of FCC. A critical next step is to identify and deliver trauma-informed integrated care and care coordination strategies that target vulnerable children and families. Increased understanding about FCC in pediatric settings that can lead to improved quality of care for children and families most at risk. As such, improved FCC is an important model of care towards advancing health equity and promoting the health and well-being of youth across the life course.

**Author Contributions:** Conceptualization, B.M.L.; methodology, B.M.L.; formal analysis, B.M.L. and H.L.; writing—original draft preparation, B.M.L., L.d.Z. and K.M.G.; writing—review and editing, L.d.Z. and H.L. All authors have read and agreed to the published version of the manuscript.

**Funding:** This research received no external funding.

**Institutional Review Board Statement:** Ethical review and approval were waived for this study due to secondary data analysis of de-identified, publicly available data.

**Informed Consent Statement:** Not applicable.

**Data Availability Statement:** The National Survey of Children's Health (NSCH) is publicly available. Data and codebook can be found electronically on the world wide web: https://www.childhealthdata.org/learn-about-the-nsch/NSCH (accessed on 16 October 2022).

**Conflicts of Interest:** The authors declare no conflict of interest.

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
