# Peer review of "Do Families Exposed to Adverse Childhood Experiences Report Family Centered Care?"

_societies, doi:10.3390/soc12060168_

Round 1

Reviewer 1 Report

The manuscript is well written and organized. As the unique contribution of the manuscript comes from the examination of ACE and its relation to FCC, I am wondering whether it is relevant to examine the types of ACE, in addition to the frequency of ACE. 

Also, the author(s) noted that prior literature clearly demonstrated the higher likelihood of ACE for children of racial/ethnic minority and low socio-economic status. With this, I think it will be important to examine the interaction effect of these variables and ACE. At minimum, I would like to see the correlation between these variables and ACE via bivariate analyses. If high correlations are reported, it is likely to impact the regression analytic model the author(s) had in this manuscript. 

Additional minor comments:

1. There are minor errors in spelling/grammar throughout the manuscript.

2. In Table 1, the percentage of FCC was 100% for the sub-sample. Is this correct? Also, in the same table, no statistics were provided for each component of FCC for the total sample.

Author Response

Please see attached document which has outlined each reviewers comments/suggestions and our related response.

Reviewer 2 Report

The article deals with the important issue of receiving family-centered care, related to the quality of care among children with adverse childhood experiences.

The article clearly presents the background, the research method, the data analysis as well as the discussion, the limitations of the research and its conclusions.

The only part that needs improvement is the findings. In its current form, all tables are presented after the description of the findings. This makes it difficult to read and understand the findings. It is recommended to present each table separately with an explanation of the relevant findings before or after it. I think this will make it easier for the reader.
